# Shunt-Induced Hepatic Encephalopathy in TIPS: Current Approaches and Clinical Challenges

**DOI:** 10.3390/jcm9113784

**Published:** 2020-11-23

**Authors:** Philipp Schindler, Hauke Heinzow, Jonel Trebicka, Moritz Wildgruber

**Affiliations:** 1Institute of Clinical Radiology, University Hospital Muenster, D-48149 Muenster, Germany; Philipp.Schindler@ukmuenster.de; 2Department of Gastroenterology and Hepatology, University Hospital Muenster, D-48149 Muenster, Germany; hauke.heinzow@ukmuenster.de; 3Section Translational Hepatology, Department for Internal Medicine I, Goethe University Frankfurt, D-60596 Frankfurt, Germany; jonel.trebicka@kgu.de; 4Department of Radiology, University Hospital, LMU Munich, D-81377 Munich, Germany

**Keywords:** hepatic encephalopathy, transjugular portosystemic intrahepatic shunt, liver cirrhosis

## Abstract

Transjugular intrahepatic portosystemic shunt (TIPS) is an established treatment tool in decompensated liver cirrhosis that has been shown to prolong transplant-free survival. Hepatic encephalopathy (HE) is a frequent complication of decompensated cirrhosis, eventually induced and/or aggravated by TIPS, that remains a clinical challenge especially in these patients. Therefore, patient selection for TIPS requires careful assessment of risk factors for HE. TIPS procedural parameters regarding stent size and invasive portosystemic pressure gradient measurements thereby have an important role. Endovascular shunt modification, in combination with a conservative medical approach, often results in a significant reduction of symptoms. This review summarizes HE molecular mechanisms and pathophysiology as well as diagnostic and therapeutic approaches targeting shunt-induced HE.

## 1. Background

Transjugular intrahepatic portosystemic shunt (TIPS) placement is an established method in the management of complications of portal hypertension [1,2]. With technical progress and increasing evidence, TIPS has improved transplant-free survival and TIPS-associated complications were vastly reduced [1,3,4,5]. However, hepatic encephalopathy (HE) occurs frequently after TIPS procedures with an incidence of 20% to 50% [6,7]. The mechanism of HE is complex, including reduced hepatic filter function in liver dysfunction and splanchnic blood shunting into the systemic circulation, as well as an overproduction of enteric neurotoxins and increased cerebral inflammation and neurotoxins [8,9,10,11,12]. Yet, hyperammonemia remains the central underlying cause [8,9,10,11,12]. The clinical effects range from mild cognitive alteration to coma, and are commonly graded using the West Haven Criteria [13,14].

Historically, post-TIPS HE pharmacological approaches are directed at reducing enteric neurotoxin production and absorption, which is increasingly being questioned today [12,15,16,17]. However, up to 8% of TIPS patients develop refractory HE, which is often associated with further deterioration of liver disease [18,19]. In these cases, endovascular shunt modification is the only therapeutic option besides liver transplantation [18,20]. TIPS modification reduces the portosystemic shunt volume and can improve HE [21].

Despite these advances, there are still uncertainties regarding the appropriate workup for TIPS patients [22]. Moreover, prevention and management of post-TIPS HE are still in need of improvement [1,22]. Correct patient selection for TIPS requires careful assessment of risk factors for HE to prevent complications which may weaken the improved hemodynamic results and worsen the patient’s life quality or expectancy [22,23,24]. Furthermore, the history of HE with increased severity itself is one of the main risk factors for HE recurrence in cirrhosis, and also an important predictor of post-TIPS HE [25,26,27].

## 2. Pathogenesis and Molecular Mechanisms

Understanding the underlying pathophysiology and molecular mechanisms of HE is essential for targeted management. Several new pathogenetic mechanisms have recently been identified while neurotoxicity from hyperammonemia remains the central underlying cause of HE [10,11,28,29].

### 2.1. Ammonia Homeostasis in Normal Liver Function and Hepatic Failure

Figure 1 illustrates the gut–liver–brain axis pathway considering porto-systemic hemodynamics. Under physiological conditions, ammonia enters the portal circulation and is cleared by the urea cycle in the liver which is then excreted by the kidneys and metabolized in skeletal muscle (Figure 1A) [10,30]. In chronic liver disease or hepatic failure, blood flow becomes hepatofugal and retrograde into the portal vein, resulting in splanchnic blood shunting into the systemic circulation (Figure 1B) [30]. Brain and muscle tissue use the enzyme glutamine synthetase to detoxify ammonia by synthesizing glutamine from glutamate [31]. The kidneys are able to release ammonia from the glutamine incurred by the brain and muscles into the urine using the enzyme glutaminase, but this mechanism is oversaturated in severe or chronic hyperammonemia [31].

### 2.2. Cerebral Ammonia Metabolism

Astrocytes are of central importance for the maintenance of adequate neuronal function and play a central role in the pathophysiology of HE [10]. They are the only central nervous system (CNS) cells capable of detoxifying ammonia by synthesizing glutamine from the excitatory neurotransmitter glutamate [32,33]. Glutamine increases the permeability of the blood-CSF barrier [33]. Acute HE, e.g., post-TIPS, is caused by a rapid rise in ammonia levels and often associated with generalized swelling of the astroglia, which clinically may present as cerebral edema (Figure 1C) [9,33,34,35]. In contrast, a long-term increase in serum ammonia levels usually does not show clinical signs of cerebral edema [9,36]. CNS cells exhibit osmotic adaptive mechanisms which may explain the lower frequency of brain edema in chronic hepatic failure [36]. Here, hyperammonemia results in direct neuronal toxicity and altered neurotransmission leading to HE (Figure 1B) [9,37].

### 2.3. Ammonia Homeostasis among TIPS Patients

Among TIPS patients, there is predominantly hepatopetal blood flow towards the low-pressure shunt rather than liver parenchyma, whereas intrahepatic portal vein flow is hepatofugal and towards the shunt (Figure 1C) [30]. Moreover, following TIPS there is an upregulation of glutaminase activity in the gut resulting in increased intestinal ammonia production [30,38]. On the other hand, the body composition can alter among TIPS patients resulting in reversal of sarcopenia and thereby improving ammonia metabolism in skeletal muscle [39,40]. However, acute post-TIPS HE is caused by a rapid short-term increase in ammonia levels and often associated with cerebral edema [9,33,34,35]. In contrast, chronic/late post-TIPS HE with a long-term rise in serum ammonia levels usually does not present as cerebral edema [9,36]. Endovascular shunt modification reduces the shunt flow and achieves hepatopetal flow reversal in the portal vein (Figure 1D) [30]. Consequently, the intestinally derived ammonia shunt is reduced and perfusion to the hepatocytes is increased, thereby improving HE [30].

### 2.4. Additional Mechanisms Underlying HE

Besides the direct correlation between blood ammonia levels and the degree of HE, further mechanisms in HE have recently been discussed [41,42]. An increased inhibitory neurotransmission by γ-aminobutyric acid (GABA) is another possible factor [11]. GABA can be formed in the colon by bacterial decarboxylation of glutamate, and, in the event of liver failure due to reduced hepatic clearance via the blood-CSF barrier, can reach the CNS [43]. The GABA receptor on the postsynaptic membrane contains binding sites for benzodiazepines [44]. By binding them, the affinity of the receptor for GABA itself is significantly increased [11,44]. This explains why benzodiazepines can trigger or worsen HE in cirrhosis of the liver [45]. It has also been shown that cirrhosis of the liver can lead to an increase in the concentration of so-called endogenous benzodiazepines [46]. A systemic inflammatory reaction with inflammatory cytokines and oxidative stress is assigned an important role [47]. Moreover, magnetic resonance imaging shows an increased deposition of manganese in the basal ganglia of patients with HE, which is neurotoxic and is usually excreted hepatobiliary [10]. In contrast to the aromatic amino acids, the branched-chain amino acids in serum are reduced in cirrhosis of the liver, since the former are degraded less in the liver and the latter are catabolized more in extrahepatic tissues [48,49]. The increased concentrations of aromatic amino acids are said to inhibit the intracerebral synthesis of dopamine and norepinephrine, while inactive false neurotransmitters are increasingly being produced [48,50]. In the colon there are toxic short- and medium-chain fatty acids as well as free, unconjugated phenols, toxic para-hydroxyphenolic acids and mercaptans (metabolites of the amino acids tyrosine, phenylalanine and methionine), which enter the CNS as a result of liver insufficiency and via portosystemic shunts, where they also have neurotoxic effects [50,51].

## 3. Epidemiology and Clinical Presentation

With a prevalence of 20–80%, HE is one of the most important comorbidities in patients with advanced cirrhosis [25,52,53]. The 1-, 5- and 10-year cumulative incidence of HE is between 0% to 21%, 5% to 25%, and 7% to 42% [25], respectively. The creation of a portosystemic shunt can significantly worsen a HE or even cause it, while the overall prevalence of HE in patients with TIPS ranges between 10% and 50% [25,53]. Within 2 years after TIPS insertion, incidence of HE has been reported between 20% and 55% [25,53].

Considering its complexity, HE can be characterized using four parameters (Table 1): (1) underlying disease, (2) severity of clinical manifestation, (3) time course, (4) existence of precipitating factors [25,54,55].
(1)According to the underlying cause, HE is subdivided into type A (due to reduced detoxification performance of the liver in acute liver failure), type B (if the hepatic detoxification function is bypassed by portosystemic bypass or shunt) and type C (by combining the mechanisms mentioned above in cirrhosis) [25,54,55].(2)Clinical effects range from mild confusion to coma and are commonly graded by the West Haven Criteria [14,54]:Grade 0 (minimal)—normal state of consciousness, objectifiable only by neuropsychiatric tests;Grade 1—slight mental slowdown, disturbed fine motor skills;Grade 2—increased fatigue, apathy, flapping tremor/asterixis, ataxia, slurred speech;Grade 3—somnolence, marked disorientation, rigor, stupor;Grade 4—coma.

However, this classification is increasingly being questioned because of its subjective nature and impaired suitability for follow-up, and has been extended by the subdivision into covert and overt HE according to the International Society for Hepatic Encephalopathy and Nitrogen Metabolism (ISHEN) guidelines [56,57].
(3)Based on its time course, HE is classified into an episodic (HE bouts more than 6 months apart), a recurrent (HE bouts within a time frame of 6 months or less) and a persistent (patterns of behavioral alterations that are always present interspersed with relapses of overt HE) form [54,55].(4)In the event of overt HE occurrence, triggering factors such as constipation, infections, gastrointestinal bleeding, electrolyte imbalance, diuretic over dosage or taking benzodiazepines, analgesics or hypnotics must always be cared for [25,54,55]. In the absence of precipitating factors, HE is considered to be spontaneous [54,55].

## 4. Risk Factors of HE in Cirrhosis and Following Tips

Despite the increased risk of developing HE or worsening it on the basis of chronic liver disease, TIPS has been well established in the therapy of refractory complications of portal hypertension such as variceal bleeding, ascites, hepatic hydrothorax and hepatorenal syndrome [1,2]. To ensure prevention and improve management of post-TIPS HE, careful patient selection for TIPS considering specific risk factors of HE is required [22,23,24]. Within this context, it is important to differentiate specific predictive risk factors of HE in cirrhosis and following TIPS placement.

### 4.1. Risk Factors in Cirrhosis

Most episodes of overt HE occur secondary to precipitating factors, especially infections and gastrointestinal bleeding [25,54]. Moreover, several studies have identified multiple risk factors of HE in cirrhosis (Table 2) [58,59,60,61,62]. Among patients with liver cirrhosis, minimal HE, history of overt HE, sarcopenia, epilepsy, diabetes, higher creatinine and bilirubin levels, lower albumin levels, and use of proton pump inhibitors and non-selective beta blockers are the main risk factors for developing overt HE [25,58,59,60,61,62]. Minimal HE is not only one of the main risk factors for overt HE, but is also associated with the severity and progression of chronic liver disease [25,63]. In a retrospective study of 216 cirrhotic patients, the occurrence of minimal HE was associated with a 2-fold increase in the risk of developing overt HE [59]. Those results were in line with the results of a later prospective study of 310 cirrhotic patients while the risk of an overt HE was 1.79 (95% CI: 1.21–2.65) compared to those without minimal HE [62]. The same studies have identified overt HE as another important risk factor for HE recurrence (2.01–2.45-fold increase) [59,62]. Moreover, the number of HE episodes also revealed a direct association with the occurrence of further HE episodes [25,64]. Changes in serum levels of albumin, bilirubin, and creatinine have also been identified as independent risk factors of HE among cirrhotic patients [25,58,59,60,61,62]. Hyperbilirubinemia and higher levels of creatinine significantly increased the risk of HE while an increase in albumin levels of 1 mg/dl reduced the risk of overt HE by up to 53% [58,59,60,61,62]. Ultimately, various comorbidities such as diabetes and epilepsy, sarcopenia and hyponatremia were assigned an increased risk of developing HE [25,58]. In addition, taking proton pump inhibitors and non-selective beta blockers also increased the risk of HE in cirrhosis (34–83%), whereas taking statins revealed a protective effect (risk reduction by 20%) [61,62].

### 4.2. Risk Factors Following TIPS

Among TIPS patients, older age, higher Child–Pugh and model of end-stage liver disease (MELD) score, history of HE pre-TIPS, low portosystemic pressure gradient (PPG), sarcopenia, and use of proton pump inhibitors were identified as potential risks for developing overt HE post-TIPS (Table 3) [25,26,65,66,67,68]. In a prospective study of 82 TIPS patients, older age was associated with an adjusted hazard ratio of 1.05 [66]. Those results were in line with the results of later retrospective studies of 284 and 264 TIPS patients [67,68]. Several studies have identified higher Child–Pugh/Child–Turcotte–Pugh and MELD score as independent risk factors of post-TIPS HE [25,26,65,66,67,68]. In a retrospective cohort of 279 TIPS patients, a unit-increase in the Child–Pugh score was associated with 1.2 higher odds of post-TIPS HE [25,65]. A prospective study of 82 TIPS patients reported similar results with an increased risk of post-TIPS HE by 29% [25,26]. Two retrospective (*n* = 279, *n* = 284) and one prospective study (*n* = 46) revealed that a one unit-increase in the MELD score was associated with 1.69 higher odds and 1.16-fold increase in the risk of post-TIPS HE, and 1.06-fold increase in the rate of new or worsening HE [25,26,65,67]. History of HE is not only one of the main risk factors for HE recurrence in cirrhosis, but also an important predictor of post-TIPS HE [25,27,66,67]. In a prospective study of 82 TIPS patients, the occurrence of HE was associated with a 3.16-fold increase in the risk of developing a post-TIPS HE [66]. A retrospective study of 284 patients reported that patients with pre-TIPS HE had 1.06-fold increase in the rate of post-TIPS HE [67]. Those results (older age, prior HE and higher Child–Pugh class/score) were similar to the results of an earlier meta-analysis of 30 studies [27]. Later studies have identified PPG as another important risk factor of post-TIPS HE [25,65]. Odds for post-TIPS HE was increased 1.2-fold for each 1 mmHg decrease in the post-TIPS PPG [65]. Comparable to the risk factors of developing HE in cirrhosis, various comorbidities such as diabetes, sarcopenia and hyponatremia were associated with an increased risk of developing post-TIPS HE [26,68]. Moreover, using proton pump inhibitors were assigned a 3.19-fold increase in the risk of post-TIPS HE [67]

## 5. Shunt Diameter and HE

The first generation of TIPS stents has been widely used, and initial underdilatation was intended to balance portal hypertension reduction and adverse events due to excessive shunting, especially HE [69]. However, several studies have shown that initial underdilatated TIPS stents tend to passively expand over time (depending on the stiffness of the liver), thereby potentially increasing HE rate [69,70,71,72]. Since 2017, so-called controlled expansion stent grafts have been established [69]. They proved to prevent self-expansion and to keep a stable shunt diameter, thereby reducing TIPS-associated complications [69,73,74]. Besides therapeutic shunting, it is well known that spontaneous portosystemic shunts have also a significant influence on HE development [75]. Spontaneous portosystemic shunts are common in cirrhotic patients and larger single shunt diameters were associated with the development of HE [76]. In a recent retrospective multicentric study of 908 cirrhotic patients, the total cross-sectional spontaneous shunt volume/area (rather than single shunt diameter) has been identified as an independent predictor of HE and survival, and should be considered for risk stratification in the work-up of cirrhotic patients [77]. Regarding shunt diameter, is has been shown that 8 mm sized covered stents do not compromise shunt function but reduce hepatic encephalopathy compared to 10 mm sized stents [78].

## 6. Management and Outcome of Post-Tips HE

Historically, patients with post-TIPS HE were conservatively treated with a low-protein diet as well as nonabsorbable antibiotics and disaccharides to reduce intestinal neurotoxin production and absorption, which is increasingly being questioned today [12,15,16,17]. Considering the limitations of current standard-of-care medications, non-pharmacological treatment strategies targeting gut dysbiosis and including probiotics and fecal microbiota transplants are increasingly used as alternative or supportive therapies [12]. Recent randomized controlled trials and meta-analyses indicated probiotics to be efficacious in treating HE compared with placebo, although there was no increased efficacy compared with lactulose [12,79,80,81]. Fecal microbiota transplantation (FMT) is increasingly established in the management of clostridium difficile infection [82]. Moreover, promising findings suggest that FMT may play an important role also in the management of other diseases associated with the disbalance of gut microbiota, including HE [82,83]. In the first randomized controlled trial, Bajaj et al. reported that FMT decreased HE recurrence, hospitalization, and improved cognitive functions compared to current standard-of-care medications (rifaximin/lactulose) [12,83]. Moreover, nutritional management and branched chain amino acids can be considered as adjunct therapies preventing degradation of skeletal muscles that detoxify ammonia [12,48,84].

However, up to 8% of patients develop refractory HE after TIPS which is often associated with further deterioration of liver function and poor prognosis [18,19,85]. In these cases, TIPS modification is increasingly integrated in multimodal treatment settings to avoid or delay liver transplantation [18,20,86,87]. Endovascular shunt modification can be performed either as partial occlusion with the insertion of a reduction stent or complete occlusion, each reducing the portosystemic shunt volume and thereby improving HE [18,87]. Within these concepts it is important to be aware of patient safety in the course of TIPS modification, particularly regarding the recurrence of the primary TIPS indication, especially variceal bleeding and ascites [19,20,86,87]. Recent main studies analyzing the effect of shunt modification in patients with refractory post-TIPS HE are summarized in Table 4 [4,19,66,87,88,89]. While shunt reduction at the time of the pivotal study by Kochar et al. was still a technical challenge, today, ready-to-use reduction stents offer the opportunity to easily downsize the shunts in a standardized manner [86,87,89]. From the data available, shunt reduction to 5 mm does not lead to relapse of variceal bleeding or refractory ascites in the majority of patients, demonstrating that reduction is mostly safe and should preferably be performed compared to complete shunt occlusion. However, endovascular shunt modification is not always successful in managing HE while possibly the presence of large collaterals and the deterioration of the liver function is more important than the changes in portal hemodynamics [38,87]. In a recent study, it has been shown that a higher HE grade after TIPS as only positive predictor for response to shunt modification, independent of liver function and PPG [89]. Non-responders reveal poor prognosis between 27–67% survival rate of 6-month follow-up following shunt modification, while liver transplantation remains the ultimate treatment [86,87,89].

## 7. Summary

Post-TIPS HE remains a clinical challenge. There are multiple factors impacting risk and prognosis of HE which have to be considered for the appropriate TIPS workup and targeted HE management. Moreover, besides the central thesis of hyperammonemia being the underlying cause of HE among TIPS patients, further molecular mechanisms have been identified and may also play an important role in the pathophysiology of HE. Current standard-of-care medications have their own limitations and non-pharmacological treatment strategies targeting gut dysbiosis may be the future of supportive therapy. In cases of refractory post-TIPS HE, endovascular shunt modification is increasingly established in multimodal treatment approaches to obviate liver transplantation. Here, TIPS reduction can be considered to be a safe treatment option that is frequently not associated with a relapse of the initial TIPS indication, such as variceal bleeding or ascites.

## Figures and Tables

**Figure 1 jcm-09-03784-f001:**
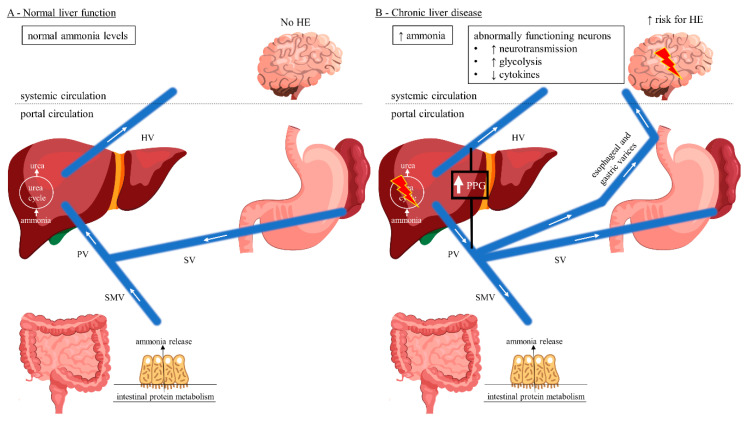
Gut–liver–brain pathway of HE considering porto-systemic hemodynamics. (**A**) Normal liver function with hepatopetal blood flow. Ammonia released from the gut enters the portal circulation and is detoxified in the liver via the urea cycle. Normal systemic ammonia levels, no HE being present. (**B**) Chronic liver disease with hepatofugal blood flow reversal resulting in development of porto-systemic collaterals and splanchnic blood shunting into the systemic circulation. Hepatic urea cycle is bypassed resulting in elevated systemic ammonia levels with increased risk for HE due to neuronal dysfunction. (**C**) TIPS with predominantly hepatopetal blood flow towards the low-pressure shunt rather than liver parenchyma, achieving decreased splanchnic blood shunting. Intrahepatic portal vein flow is hepatofugal and into the shunt. The hepatic urea cycle is again bypassed resulting in elevated systemic ammonia levels with increased risk for HE, in an acute setting, due to the increased metabolism of ammonia to glutamine by the astrocytes, with a subsequent tendency to edema. (**D**) Endovascular shunt modification decreases the shunt flow achieving hepatopetal flow reversal in the portal vein. Ammonia detoxification in the liver via the urea cycle is increased, thereby improving HE. This cover was created with resources from Freepik.com. Abbreviations: HE, hepatic encephalopathy; HV, hepatic vein; PPG, portosystemic pressure gradient; PV, portal vein; SMV, superior mesenteric vein; SV, splenic vein; TIPS, transjugular intrahepatic portosystemic shunt.

**Table 1 jcm-09-03784-t001:** Clinical characterization of HE (adapted from AASLD/EASL guidelines [54]).

Type	Grade		Time Course	Spontaneous/Precipitated
A	Minimal	Covert	Episodic	Spontaneous
	I			
B	II		Recurrent	
	III	Overt		Precipitated factors
C	IV		Persistent	

**Table 2 jcm-09-03784-t002:** Recent relevant publications identifying specific risk factors of overt HE in cirrhosis (adapted from Elsaid et al. [25]).

Reference	Study Design	Number of Patients	Risk Factor	Adjusted Hazard Ratio (95% CI)
Jepsen et al., 2015 [58]	Secondary analysis *	862	Diabetes	1.86 (1.20–2.87)
			Child–Pugh class B	2.57 (0.61–10.8)
			Child–Pugh class C	4.32 (0.96–19.3)
			Bilirubin, per 10 µmol/l increase	1.06 (1.03–1.08)
			Albumin, per 5 g/L increase	0.68 (0.56–0.83)
			Sodium, per 5 mmol/L increase	0.63 (0.53–0.74)
			Creatinine, per 10 μmol/L increase	1.09 (1.05–1.13)
Riggio et al., 2015 [59]	Retrospective cohort	216	Previous overt HE	2.01 (1.24–3.26)
			Minimal HE	2.02 (1.23–3.33)
			Albumin level < 3.5 g/dL	2.32 (1.37–3.93)
Ruiz-Margáin et al., 2016 [60]	Prospective cohort	220	Cachexia	1.81 (1.08–3.03)
			Creatinine, per 1 mg/dL increase	4.12 (1.57–10.77)
Tapper et al., 2018 [61]	Retrospective cohort	1979	Bilirubin, per 1 mg/dL increase	1.07 (1.05–1.09)
			Albumin, per 1 mg/dL increase	0.54 (0.49–0.60)
			Non-selective beta-blocker use	1.34 (1.09–1.64)
			Statin use	0.80 (0.65–0.98)
Nardelli et al., 2019 [62]	Prospective cohort	310	Albumin, per 1 g/L increase	0.47 (0.33–0.69)
			Previous overt HE	2.45 (1.66–3.58)
			Minimal HE	1.79 (1.21–2.65)
			Proton pump inhibitors use	1.83 (1.22–2.74)

* data from 3 randomized trials; Abbreviations: CI, confidence interval; HE, hepatic encephalopathy.

**Table 3 jcm-09-03784-t003:** Recent relevant publications identifying specific risk factors of post-TIPS HE.

Reference	Study Design	Number of Patients	Risk Factor	Adjusted Hazard Ratio (95% CI)
Yao et al., 2015 [65]	Retrospective cohort	279	Pre-TIPS MELD	1.69 (1.39–2.06) ^+^
			PPG post-TIPS	1.20 (1.07–1.34) ^+^
Nardelli et al., 2016 [66]	Prospective cohort	82	Age	1.05 (1.02–1.08)
			Child–Pugh score	1.29 (1.06–1.56)
			Covert HE before TIPS	3.16 (1.43–6.99)
Nardelli et al., 2017 [26]	Prospective cohort	46	Sarcopenia	31.3 (4.5–218.07)
			Pre-TIPS MELD	1.16 (1.01–1.34)
Lewis et al., 2019 [67]	Retrospective cohort	284	Age	1.05 (1.03–1.07) *
			Pre-TIPS MELD	1.06 (1.01–1.11) *
			HE before TIPS	1.51 (1.04–2.20) *
			Proton pump inhibitors use	3.19 (2.19–4.66) *
Yin et al., 2020 [68]	Retrospective cohort	264	Age	1.03 (1.00–3.21)
			Diabetes	1.84 (1.06–3.21)
			Child-Turcotte-Pugh class C	6.68 (1.68–8.89)
			Sodium	0.94 (0.88–0.99)
			Creatinine	1.01 (1.00–1.03)

^+^ = odds ratio; * IRR = incidence rate ratio; Abbreviations: CI, confidence interval; HE, hepatic encephalopathy; MELD, model of end-stage liver disease; PPG, portosystemic pressure gradient, TIPS, transjugular intrahepatic portosystemic shunt.

**Table 4 jcm-09-03784-t004:** Recent relevant publications analyzing the effect of shunt modification in patients with refractory post-TIPS HE (adapted from Nardelli et al. [87]).

Reference	No. with Refractory HE/Treated with TIPS	Child–Pugh Class	No. of Patients Improved	Recurrence of Primary TIPS Indication after Shunt Modification	PPG Pre (mmHg)	PPG Post (mmHg)
Nardelli et al., 2016 [66]	3/82	B: 1 C: 2	3	-	5.6 ± 3.2	12.1 ± 2.7
De Santis et al., 2018 [88]	2/38	B: 1 C: 1	2	Ascites 1 Bleeding 1	6.5 ± 2.6	12.7 ± 3.8
Bureau et al., 2017 [4]	1/29	C: 1	1	-	-	-
Rowley et al., 2018 [19]	10/174	-	8	-	8.6 ± 4.1	13.0 ± 4.0
Schindler et al., 2020 [89]	20/344	A: 7 B: 9 C: 4	11	Ascites 2 Bleeding 1	7.7 ± 3.9	12.1 ± 4.4

- = not calculated or specified; Abbreviations: HE, hepatic encephalopathy; PPG, portosystemic pressure gradient, TIPS, transjugular intrahepatic portosystemic shunt.

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
