# Peer review of "Shunt-Induced Hepatic Encephalopathy in TIPS: Current Approaches and Clinical Challenges"

_jcm, 2020, doi:10.3390/jcm9113784_

Round 1

Reviewer 1 Report

Philipp Schindler et al. wrote a review entitled: Shunt-induced Hepatic Encephalopathy in TIPS: current approaches and clinical challenges.

The review is well written and comprehensive and covers all important aspect of HE and TIPS placement.  

The ”PATHOGENESIS AND MOLECULAR MECHANISMS” part is long. For most researchers within the HE area, most of this is well known and maybe the text could be shortened a bit. The patient selection for TIPS and the assessment of risk factors for HE is the most interesting part and could be expanded.

A proper reference for brain edema and HE could be added. There are several, both review and MRI studies.

Line 116: Could consider adding a reference which describes the relevant BCAA and ammonia fluxes across muscle: Example:

Branched-chain amino acids increase arterial blood ammonia in spite of enhanced intrinsic muscle ammonia metabolism in patients with cirrhosis and healthy subjects.

Dam G, Keiding S, Munk OL, Ott P, Buhl M, Vilstrup H, Bak LK, Waagepetersen HS, Schousboe A, Møller N, Sørensen M.Am J Physiol Gastrointest Liver Physiol. 2011 Aug;301(2):G269-77. doi: 10.1152/ajpgi.00062.2011. Epub 2011 Jun 2. But many others are also relevant. 

Figure 1. Gut-liver-brain pathway of HE considering porto-systemic hemodynamics: The blue blood flow lines: The graphics should be improved a bit.

Reviewer 2 Report

I wish to congratulate the Authors of the manuscript.

The article is well structured, the authors discuss the aspects of hepatic encephalopathy clearly and in a logical sequence.

The authors provide conclusions from studies published up to five years that makes this work actual.

I am not an English-native speaker, thus linguistic correctness assessment is beyond my competence. Nevertheless, the article was perfectly understandable by me and I couldn't have written it better.

Reviewer 3 Report

The article is very well written and reviews all the important points about TIPS and encephalopathy. 

There are only some minor questions:

  • Regarding the causes of encephalopathy in patients with cirrhosis I think it would be important to talk about constipation and therapy with diuretics as they are one of the main attributable causes for encephalopathy in our daily practice.
  • Did you find any information if the reason for placing TIPS (variceal bleeding, ascites) could influence the risk of encephalopathy after TIPS? 
  • What was the best diameter for TIPS to reduce the risk of encephalopathy?
  • Are there any information about FMT after TIPS?
  • In page number 5, I would suggest to remove the text of EASL/AASLD guidelines and presented only in boxes as figures.
